# Translation and Fusion Improves Zero-shot Cross-lingual Information Extraction

## Abstract

Large language models (LLMs) combined with instruction tuning have shown significant progress in information extraction (IE) tasks, exhibiting strong generalization capabilities to unseen datasets by following annotation guidelines. However, their applicability to low-resource languages remains limited due to lack of both labeled data for fine-tuning, and unlabeled text for pre-training. In this paper, we propose TransFusion, a framework in which models are fine-tuned to use English translations of low-resource language data, enabling more precise predictions through annotation fusion. Based on TransFusion, we introduce GoLLIE-TF, a cross-lingual instruction-tuned LLM for IE tasks, designed to close the performance gap between high and low-resource languages. Our experiments across twelve multilingual IE datasets spanning 50 languages demonstrate that GoLLIE-TF achieves better cross-lingual transfer over the base model. In addition, we show that TransFusion significantly improves low-resource language named entity recognition when applied to proprietary models such as GPT-4 (+5 F1) with a prompting approach, or fine-tuning different language models including decoder-only (+14 F1) and encoder-only (+13 F1) architectures.

## 1 Introduction

The task of information extraction (IE) is challenging due to fine-grained annotation guidelines for span-level annotations. Fortunately, recent advances in instruction-following large language models (LLM) (Ouyang et al., 2022; Gemini et al., 2023) such as GoLLIE (Sainz et al., 2024) have demonstrated the ability to perform zero-shot IE without labels using annotation guidelines. However, these models are often pre-trained on English-centric data (Touvron et al., 2023; Roziere et al., 2023). Even state-of-the-art proprietary models such as GPT-4 exhibit significant performance degradation from 80 English F1 to 55 F1 on low-resource African languages, as shown in Figure 1 (right).

To improve NLP on low-resource languages, the research community has turned to machine translation to translate fine-tuning datasets (translate-train) and translate test data into high-resource languages for easier processing (translate-test) (Hu et al., 2020). Recent studies (Shi et al., 2022; Huang et al., 2023) on prompting LLMs with translated data have shown improvements on diverse tasks such as math reasoning and summarization. Prior work has explored the use of machine translation to improve multilingual instruction-following on traditional NLP benchmarks, such as natural language inference, and sentiment analysis, however, the use of MT to improve instruction-following IE models is less explored, as there is not a trivial alignment between labels in the native language and translated texts (Ahuja et al., 2023). With recent efforts to develop machine translation (MT) models such as M2M (Fan et al., 2021) and NLLB-200 (Costa-jussà et al., 2022) that better support low-resource languages, we study how to teach LLMs to leverage an external MT system in a resource-efficient manner to improve low-resource IE.

In this paper, we propose a Translation and Fusion (TransFusion) framework, which aims to teach models to use translation data from an external MT system to make better predictions. The framework includes three steps: (1) translating low-resource data into English at inference time, to be annotated by a high-resource model. Next, (2) these span-annotated English translations are combined with low-resource language text in a fusion model that is trained to make predictions conditioned on both types of data. Finally (3), the language model generates a TransFusion reasoning chain (annotate and fuse) in a single autoregressive decoding pass. To train TransFusion models, we construct cross-lingual instruction fine-tuning data by translating and projecting labels from English IE datasets

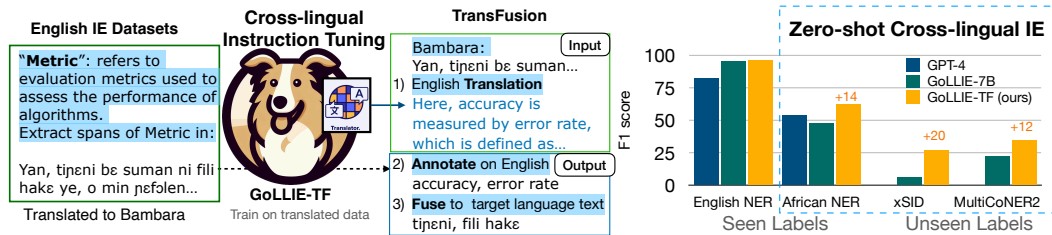

Figure 1: Our TransFusion framework aims to bridge the performance gap between high and low-resource languages on information extraction tasks. (left) TransFusion reasoning includes three steps: translate, annotate, and fuse. (right) GoLLIE-TF shows superior cross-lingual evaluation on a range of IE datasets (including unseen labels) over the base model.

to low-resource languages using EasyProject (Chen et al., 2023b), a simple, yet effective method that has been shown to scale across many NLP tasks and languages.

Our cross-lingual IE evaluation reveals that the TransFusion fine-tuned model, GoLLIE-TF, outperforms the base GoLLIE model across 50 languages, spanning high, mid, and low-resource categories, on both seen and unseen label schemas. Notably, in our evaluation on African language named entity recognition (NER) using the MasakhaNER2 dataset (Adelani et al., 2022), GoLLIE-TF achieves significant improvements in $F_1$ scores compared to the base model and shows an average improvement of +6.6 $F_1$ on unseen label schema datasets. Furthermore, we demonstrate that the TransFusion framework enhances GPT-4's performance on MasakhaNER2, yielding an average +5.7 $F_1$ score improvement, and substantially boosts the encoder-only African language model, AfroXLM-R (Alabi et al., 2022), by +13.3 $F_1$. Our analysis underscores the effectiveness of the TransFusion framework for low-resource language tasks.

## 2 BACKGROUND: ANNOTATION GUIDELINE FOLLOWING LLMS FOR IE

In this paper, we employ the GoLLIE model (Sainz et al., 2024), which has been instruction-tuned on English Information Extraction (IE) tasks using label schema guidelines, to achieve state-of-the-art zero-shot IE on unseen datasets. GoLLIE utilizes a Python code representation for both inputs and outputs, providing a clear and human-readable structure that unifies various IE annotation tasks. Each label schema is encapsulated as a Python class object, with the annotation guidelines embedded as strings within these objects (an example of a GoLLIE prompt is provided in the the Appendix in Figure 6.

**Limitation of Cross-lingual Transferbilitiy**: Despite GoLLIE's impressive performance, it is designed for use on English, as it is primarily fine-tuned on English data. This limitation is shown in Figure 1 (right), where we see a significant drop in performance on low-resource African languages, from 95 to 48, compared to English. In this study, we experiment with **cross-lingual transfer**, where human-labeled data in the target languages are assumed to be unavailable. Collecting such data is costly and time-inefficient, as it requires well-trained native language speakers. Recent efforts, such as NLLB-200 (Costa-jussà et al., 2022), have focused on gathering low-resource translation data to train multilingual MT models capable of translating across 200 languages. Building on this progress, we explore whether an instruction-tuned information extraction model can learn to use an external translation model (Schick et al., 2024) to enhance performance on low-resource language IE tasks. This offers an efficient and effective alternative to computationally intensive pre-training based methods for adapting to new languages (Scao et al., 2022; Xue et al., 2021; Alabi et al., 2022; Üstün et al., 2024).

## 3 USING LOW-RESOURCE MACHINE TRANSLATION TO IMPROVE MULTILINGUAL IE

As multilingual machine translation (MT) systems, such as M2M-100 (Fan et al., 2021) and NLLB-200 (Costa-jussà et al., 2022), gain increasing support for low-resource languages, an opportunity emerges to re-evaluate the utilization of MT systems for enhancing cross-lingual IE. We propose a Translation-and-fusion approach that benefits from the advancements of MT systems to make robust

cross-lingual transfer predictions at inference time. In this section, we outline the Translation-and-fusion framework and introduce language models trained to utilize translation data at inference time for low-resource language IE tasks.

### 3.1 TRANSLATION-AND-FUSION (TRANSFUSION)

**Cross-lingual Transfer.** The conventional cross-lingual transfer method involves fine-tuning a pre-trained language model, on high-resource language annotated data ($src$) and evaluating its performance on test data in other languages ($tgt$).

In accordance with the low-resource assumption, we assume access to an annotated dataset in the high-resource language (usually English), $\mathcal{D}_{src} = (x_{src}^i, y_{src}^i)_{i=1}^N$. The task-specific fine-tuning loss is formulated as:

$$\mathcal{L}(\theta, \mathcal{D}_{src}) = \sum_{(x_{src}, y_{src}) \in \mathcal{D}_{src}} \mathcal{L}(P(y|x_{src}; \theta), y_{src})$$

However, previous studies have highlighted the limited performance of fine-tuned models on languages that were unseen during pre-training or are under-represented in the pre-training data (Adelani et al., 2021; Ebrahimi et al., 2022). As an additional approach to adapt to low-resource languages (Wang et al., 2020), we describe the translation-and-fusion framework, which leverages annotations on (translated) high-resource language text to steer predictions on a low-resource language at inference time. The framework encompasses three key steps:

- **Translate**: Use an MT system to translate low-resource language test data into a high-resource language, $\text{MT}(x_{tgt}) \mapsto x_{src}^{\text{trans}}$.

- **Annotate**: Making predictions to the (high-resource) translated text using a high-resource language supervised fine-tuned model $P(; \theta_{src})$: $\text{argmax}_y\{P(y|x_{src}^{\text{trans}}; \theta_{src})\} \mapsto \tilde{y}_{src}^{\text{trans}}$.

- **Fuse**:

  Given the annotations on the translated data from the previous step ($\tilde{y}_{src}^{\text{trans}}$), a fusion model combines the *high-resource predictions* together with the target language text to make final predictions.

Based on the framework outlined above, we present TransFusion, a fusion model that is trained to makes predictions on the test data conditioned on annotations from the corresponding translated data ($\tilde{y}_{src}^{\text{trans}}$):

$$\text{argmax}_y\{P(y|x_{tgt}, x_{src}^{\text{trans}}, \tilde{y}_{src}^{\text{trans}}; \theta_{\text{fusion}})\} \mapsto y_{tgt}'$$

Below, we describe the training procedure of TransFusion, starting with the approach to create training data.

**Training Dataset.** To learn a TransFusion model, parallel sentences with IE task annotations on both high-resource and low-resource languages are essential. To fulfill this requirement, we translate high-resource annotated training data into a list of target languages, while projecting span-level annotations, using a simple mark-then-translate approach - EasyProject (Chen et al., 2023b): $\text{MT}(x_{src}, y_{src}) \rightarrow (x_{tgt}^{\text{trans}}, y_{tgt}^{\text{trans}})$. We then pair the translation outputs with the original high-resource language data to create a training data set with a mixture of both parallel sentences: $\mathcal{D}_{mix} = \{x_{src}, y_{src}, x_{tgt}^{\text{trans}}, y_{tgt}^{\text{trans}}\}_{i=1}^N$.

**Learning.** We train the fusion model $P(; \theta_{\text{fusion}})$ on $\mathcal{D}_{mix}$ using cross-entropy loss:

$$\mathcal{L}_{\text{fusion}}(\theta, \mathcal{D}_{mix}) = \sum_{(x_{src}, y_{src}, x_{tgt}^{\text{trans}}, y_{tgt}^{\text{trans}}) \in \mathcal{D}_{mix}} \mathcal{L}(P(y|x_{tgt}^{\text{trans}}, x_{src}, y_{src}; \theta_{\text{fusion}}), y_{tgt}^{\text{trans}})$$

The model architecture can vary, encompassing both decoder-only language models (e.g., LLaMA (Touvron et al., 2023)) and encoder-only language models (e.g., mBERT (Devlin et al., 2019)). In this work, we primarily utilize decoder-only language models to integrate the *annotate* and *fuse* steps in an autoregressive manner during inference. Additionally, we assess the performance of encoder-only models in Section 5.3 to demonstrate the robustness of our framework across different architectures.

**Training a Decoder-only LM (GoLLIE-TF).** To implement our TransFusion framework within the instruction-following GoLLIE model, we represent the framework as natural language instructions, providing the model with supplementary English translation text of the original target language sentence, which is illustrated in Figure 1 (left). The TransFusion instruction specifies the output format, guiding the model to first generate annotations for the English translation and subsequently for the target language data, using the English annotations as context (an example can be found in Appendix Figure 6 ). This autoregressive approach enables the model to perform the annotate and fuse steps concurrently during inference. During training, we fine-tune the GoLLIE model to adhere to these instructions, ensuring it generates annotations for both the English and target language data sequentially. We apply the next token prediction loss to the tokens following the TransFusion instruction. At inference time, $x$ is the low-resource language and $x^{trans}$ is the English translation:

$$[\texttt{GoLLIE Guidelines}, x, x^{trans}, \texttt{TransFusion Instruction}] \xrightarrow{\text{LLM}} [y^{trans}, y]$$

**Training and Inference with Encoder-only LMs.** Given that encoder-only models are not inherently designed for text generation, we employ a two-step pipeline approach for inference in TransFusion: annotation and fusion. First, we utilize an English fine-tuned model to annotate the English translation of the target language text. These annotations are marked using XML tags around the relevant spans (e.g., `<PER>` ... `</PER>`). Next, we construct the input for the fusion model by embedding these annotations into the English translation. We concatenate the annotated English translation ($x^{trans}$) with the original target language text ($x$), using a marker ($||$) to separate the two segments. The input to the encoder is formatted as follows:

$$[x_1^{trans}, x_2^{trans}, \texttt{<PER>}, x_3^{trans}, x_4^{trans}, \texttt{</PER>}, x_5^{trans}, ||, x_1, x_2, x_3, ...]$$

At training time, we add a linear classification layer to classify each token and only apply the cross-entropy loss to the target language tokens (right of the separation token $||$).

To summarize, Translation-and-Fusion framework can be adapted into three different configurations for different usages including decoder-only (§ 5.1), prompting (§ 5.2), and encoder-only (§ 5.3), with the same appraoch.

## 4 EXPERIMENTAL SETTING

We use a collection of English Information Extraction (IE) datasets for supervised fine-tuning and multilingual IE datasets for evaluation (see Table 1). Assessing cross-lingual transfer capabilities requires IE datasets annotated in a diverse set of languages. To this end, we gather multilingual Named Entity Recognition (NER) datasets from MasakhaNER2.0 (Adelani et al., 2022) (20 African languages) and UNER (Mayhew et al., 2023) (13 languages) to conduct low-resource language evaluation on label schemas that are seen during fine-tuning. In addition, we evaluate on unseen label schemas using the non-English subset of ACE2005 (Tjong Kim Sang & De Meulder, 2003) (Chinese and Arabic), which includes several tasks: NER, RE, Event Extraction (EE), and Event Argument Extraction (EAE). For evaluation on labels that were unseen during fine-tuning, we use MultiN-ERD (Tedeschi & Navigli, 2022) (10 high-resource languages), MultiCoNER2 (12 high-resource languages) (Fetahu et al., 2023), in addition to Slot Intent Detection data from MultiTO (Schuster et al., 2018), xSID (10 high-resource languages) (van der Goot et al., 2021), a subset of Massive (15 low-resource languages were determined based on the NLLB categorization (Costa-jussà et al., 2022)) (FitzGerald et al., 2022) and Relation Extraction (RE) data from RED-FM (7 high-resource languages) (Cabot et al., 2023). We adopt the data pre-processing and task formulation methodologies used by GoLLIE and use publicly available English training data from GoLLIE to train the model. Due to the high cost of inference with GPT-4, we use 200 examples per language (Le et al., 2024), per task, for evaluation.

**Multilingual Translation Data.** The TransFusion framework relies on a machine translation system as a core component. In this paper, we utilize the state-of-the-art open-source multilingual translation model - NLLB-200 (Costa-jussà et al., 2022), which has 3.3 billion parameters and supports translation between 200 languages. The NLLB-200-3.3B model translates target language test data into English at test time. For TransFusion training data, a marker-based translation approach named EasyProject (Chen et al., 2023b), powered by the NLLB-200 model, translates English training data into a collection of 36 target language candidates. From this translated data, 8 examples per

Table 1: Datasets used in the experiment. The table shows the task, domain, whether it was used in the training and evaluation including the number of languages in the evaluation set.

| Training Dataset | Domain | Tasks | Language |
|---|---|---|---|
| CoNLL 03(Tjong Kim Sang & De Meulder, 2003) | News | NER | English |
| BC5CDR (Li et al., 2016) | Biomedical | NER | English |
| NCBIDisease (Dogan et al., 2014) | Biomedical | NER | English |
| OntoNotes 5 (Pradhan et al., 2013) | News | NER | English |
| WNUT 2017 (Derczynski et al., 2017) | News | NER | English |
| RAMS (Ebner et al., 2020) | News | Arg. Extraction | English |
| TACRED (Zhang et al., 2017) | News | Slot Filling | English |
| CoNLL 04 (Roth & Yih, 2004) | News | Relation Extraction | English |
| ACE (Walker et al., 2006) | News | EE, EAE, NER, RE | English |

| Evaluation Dataset | Domain | Tasks | Seen Label? | # Language |
|---|---|---|---|---|
| MasakhaNER2.0 (Adelani et al., 2022) | News | NER | ✓ | 20 African langs |
| UNER (Mayhew et al., 2023) | News | NER | ✓ | 13 langs |
| ACE (Walker et al., 2006) | News | EE, EAE, NER, RE | ✓ | 3 (en, ar, zh) |
| MultiNERD (Tedeschi & Navigli, 2022) | Wikipedia | NER | ✗ | 10 langs |
| MultiCoNER2 (Fetahu et al., 2023) | Wikipedia | NER | ✗ | 12 langs |
| xSID (van der Goot et al., 2021) | Dialog | Slot Detection | ✗ | 10 langs |
| MultiTO (Schuster et al., 2018) | Dialog | Slot Detection | ✗ | 3 (en, es, th) |
| Massive (FitzGerald et al., 2022) | Dialog | Slot Detection | ✗ | 15 low-res langs |
| RED-FM (Cabot et al., 2023) | Wikipedia | Relation Extraction | ✗ | 7 langs |

language and each task are randomly sampled, resulting in around 20-40 examples per language. To summarize, we started from the GoLLIE-7B checkpoint and fine-tune the model on 20,000 examples including English (19,109) and translated data (891) (See per task and per language distribution in Appendix Figure 8). This small portion of translation data (Shaham et al., 2024) ensures that the GoLLIE model generalizes to unseen labels while maintaining English performance to avoid the catastrophic forgetting issue during continue fine-tuning (Luo et al., 2023).

### 4.1 LANGUAGE MODELS AND BASELINES

**Models:** We adopt GoLLIE-7B as our primary starting checkpoint. GoLLIE is an instruction fine-tuned version of CodeLLaMA (Roziere et al., 2023) that is trained on approximately 500,000 English demonstrations. Although the model was not explicitly pre-trained on multilingual data, its pre-training corpus includes a substantial amount of high-resource language content, such as Wikipedia, covering a diverse linguistic range (Touvron et al., 2023). This makes GoLLIE-7B an appropriate testbed for examining the adaptation of English-centric LLMs to low-resource languages that may be underrepresented in pre-training. In addition to this decoder-only LLM, we explore encoder-only models specifically pre-trained on African languages, such as AfroXLM-R (Alabi et al., 2022) in Section 5.3.

**Training Setup:** Initilized from GoLLIE-7B, we continue fine-tuning the model on a dataset of 20,000 TransFusion training examples using QLoRA (Dettmers et al., 2024). QLoRA has been shown to better maintain the base model's performance (Biderman et al., 2024) and offers faster training times compared to full fine-tuning. To implement this, we freeze the transformer model weights and apply LoRA (Hu et al., 2021) to all linear layers within all the transformer blocks. We set the LoRA rank to 128 and the alpha parameter to 16 based on preliminary experiments as we found smaller alpha leads to more stable training and higher rank for fast convergence. We use the AdamW optimizer (Kingma & Ba, 2015) with a batch size of 16 and a learning rate of 1e-4, managed by a cosine scheduler. The training process was conducted on a setup of 2 NVIDIA A40 GPUs, each equipped with 48GB of memory. The entire experiment session spanned approximately 6 hours. We use greedy decoding at inference time.

**Baselines:** We compare to both the base GoLLIE model, in addition to GPT-4, which represents a state-of-the-art proprietary model pre-trained on multilingual corpora (Achiam et al., 2023). We report

Table 2: **Cross-lingual transfer** performance (F1 score). The table compiles all the seen label schema and unseen label schema evaluation results. Blue numbers highlight the performance improvements over GoLLIE-7B ($\Delta$). Full results for each language can be found in Appendix.

| Task | Benchmark | GPT-4 | GoLLIE$_{7B}$ | Trans-Train | GoLLIE-TF |
|---|---|---|---|---|---|
| **Seen Label Schema** | | | | | |
| NER | MasakhaNER2 (20 languages) | | | | |
| | Bambara | 42.2 | 38.9 | 40.1 | **54.8** (+15.9) |
| | Ghomala | **58.2** | 43.7 | 49.2 | 50.2 (+6.5) |
| | Ewe | 72.2 | **74.0** | 73.1 | 73.2 (-0.8) |
| | Fon | 39.4 | 49.7 | 55.7 | **57.9** (+8.2) |
| | Hausa | 65.9 | 57.1 | 55.6 | **67.1** (+10.0) |
| | Igbo | 42.2 | 51.1 | 42.4 | **56.6** (+5.5) |
| | Kinyarwanda | 47.5 | 45.0 | 47.7 | **58.5** (+13.6) |
| | Luganda | 62.5 | 61.8 | 66.8 | **75.5** (+13.7) |
| | Luo | 47.2 | 36.5 | 42.8 | **51.7** (+15.3) |
| | Mossi | 43.2 | 45.1 | 46.1 | **48.8** (+3.7) |
| | Chichewa | 71.1 | 39.1 | 59.8 | **78.2** (+39.1) |
| | Naija | 78.9 | 75.9 | 74.9 | **81.1** (+5.2) |
| | Shona | 39.5 | 39.7 | 50.4 | **57.4** (+17.6) |
| | Swahili | **79.2** | 66.9 | 68.3 | 73.5 (+6.5) |
| | Tswana | 56.3 | 52.1 | 58.9 | **71.0** (+18.9) |
| | Twi | 44.2 | 41.7 | 50.6 | **74.2** (+32.5) |
| | Wolof | 52.6 | 49.1 | 55.5 | **61.9** (+12.8) |
| | Xhosa | 49.8 | 29.2 | 47.6 | **49.9** (+20.7) |
| | Yoruba | **54.7** | 35.7 | 39.3 | 54.4 (+18.7) |
| | Zulu | 36.9 | 25.6 | 31.7 | **52.8** (+27.2) |
| | Average | 54.2 | 47.9 | 52.8 | **62.4** (+14.5) |
| NER | UNER (13 languages) | 69.0 | 73.6 | 73.6 | **77.8** (+4.2) |
| NER | ACE05 (English, Arabic, Chinese) | 41.6 | 58.7 | 61.2 | **61.5** (+2.8) |
| Arg. Extraction | ACE05 (English, Arabic, Chinese) | 11.7 | 92.7 | **92.9** | 86.0 (-6.7) |
| Event Detection | ACE05 (English, Arabic, Chinese) | 21.3 | 42.6 | 40.0 | **44.0** (+1.4) |
| Rel. Extraction | ACE05 (English, Arabic, Chinese) | 4.6 | 37.3 | **39.4** | 39.1 (+1.8) |
| **Unseen Label Schema** | | | | | |
| NER | MultiNERD (10 languages) | **71.9** | 62.2 | 63.9 | 63.0 (+0.8) |
| NER | MultiCoNER2 (12 languages) | **46.1** | 22.2 | 28.4 | 34.5 (+12.2) |
| Slot Detection | xSID (10 languages) | **47.0** | 6.0 | 27.1 | 26.4 (+20.4) |
| Slot Detection | MultiTO (English, Spanish, Thai) | 19.9 | 17.7 | **20.3** | 18.1 (+0.4) |
| Slot Detection | Massive (15 low-resource languages) | **33.3** | 5.8 | 12.1 | 19.0 (+13.1) |
| Rel. Extraction | REDFM (7 languages) | **19.1** | 15.5 | 16.8 | 16.2 (+0.7) |
| | Seen | 33.7 | 58.8 | 60.0 | **61.8** (+3.0) |
| Average | Unseen | **39.5** | 21.6 | 28.1 | 29.5 (+8.0) |
| | English-only | 55.2 | 58.6 | **60.3** | 59.3 (+0.7) |
| | All | 36.6 | 40.2 | 44.1 | **45.7** (+5.5) |

few-shot prompting results using GPT-4 (`gpt4-02-14`) with a GoLLIE style prompt. Additionally, we explore the application of the TransFusion framework to GPT-4 in Section 5.2. Furthermore, we use Translate-train (**Trans-train**) (Hu et al., 2020) as another baseline, which shows strong improvements over English fine-tuned (English FT) models (Chen et al., 2023b). We use the same translated training data used by TransFusion and fine-tune GoLLIE-7B on a total of 20,000 examples (English + translated data). So the only differences between Trans-Train and GoLLIE-TF is the Trans-Train fine-tune on the $(x^{trans}, y^{trans})$ translated pairs where GoLLIE-TF is fine-tune on the four-way parallel data $(x, y, x^{trans}, y^{trans})$ with TransFusion instruction.

## 5 RESULTS

We present cross-lingual transfer results for IE tasks in Table 2, evaluating both seen and unseen label schemas across 36 languages. Our proposed GoLLIE-TF model consistently outperforms the original

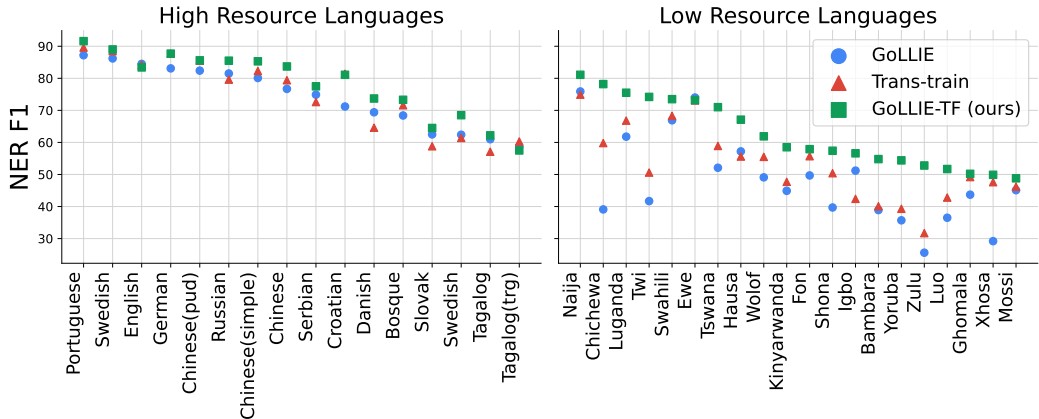

Figure 2: TransFusion leads to larger NER F1 improvements for low resource languages in MasakhaNER2 (right) compared to high resource languages in UNER (left).

GoLLIE, achieving an average F1 score improvement of +4.6 across 11 datasets. Notably, GoLLIE-TF demonstrates significant performance gains in low-resource language NER while mainting English performance on average. For instance, on the MasakhaNER2 dataset, TransFusion boosts F1 from 47.9 to 62.4, surpassing both GPT-4 and the translate-train baseline. Furthermore, GoLLIE-TF supports generalization to unseen label schemas. In particular, TransFusion significantly improves performance on MultiCoNER2 (+12.2), xSID (+20.4), and on low-resource language dataset Massive (+13.1) over GoLLIE, showcasing its adaptability to unseen tasks. While GPT-4 still demonstrates superior performance on unseen label schemas, we would like to highlight that our experiments are conducted in a controlled setting. In contrast, for proprietary models, we are unaware of the dataset used, leading to potential dataset contamination.

**TransFusion performance on High vs. Low-resource languages.** Figure 2 reveals a noteworthy trend: GoLLIE-TF exhibits substantial performance enhancements particularly in low-resource language settings. This underscores the significance of leveraging external Machine Translation systems to enrich input data for such languages. We followed the categorization of high and low-resource languages from Costa-jussà et al. (2022), which categorizes a language as low-resource if there are fewer than 1M publicly available deduplicated bitext samples. While the performance disparity between GoLLIE-TF and other models remains modest in high-resource language scenarios, a notable performance gap emerges in the low-resource language domain. Furthermore, results on the unseen-label low-resource language dataset, Massive, also show that GoLLIE-TF signficiantly outperforms Translate-Train, as shown in in Table 2.

### 5.1 ABLATION STUDY

**Analyzing Performance Improvements** Table 3 shows a critical insight into the performance gains observed in the TransFusion framework, particularly in the *annotate* step on the English translation, which plays a crucial role in enhancing the performance of MasakhaNER2. We conduct an ablation study wherein we trained a variant of GoLLIE-TF, termed GoLLIE-TF (w/o *annotate*), directly generating predictions on target language text from the unlabelled

Table 3: Ablation study.

| Model | MasakhaNER2 | MASSIVE |
|---|---|---|
| GoLLIE-TF | 62.4 | 19.0 |
| - w/o annotate | 55.7 | 13.3 |
| - no translation | 41.2 | 10.7 |

English text. We observe a notable performance drop from 62.4 to 55.7 F1 score. This observation underscores the significance of TransFusion's ability to leverage English annotations during test time, resulting in more precise predictions. Furthermore, we take the GoLLIE-TF model to direct make inference on target language without translation (*no translation*), the performance further drops to 41.2 and 10.7 on MasakhaNER2 and MASSIVE, showing the importance of using translation data at the test time.

**Effectiveness at different training data size.** In Table 4, we explored the impact of varying the amount of translated data (ranging from 1000 to 40000) combined with 19000 English data for

training. The results demonstrate that across all scales, GoLLIE-TF consistently outperforms the trans-train baseline on the MasakhaNER task, with performance improving from 62.4 to 66.3 as the translation data size increases from 1000 to 40000, compared to trans-train's performance increase from 52.8 to 56.4. These results highlight the effectiveness of GoLLIE-TF in leveraging both English and translated data for improved NER performance.

Table 4: NER performance on MasakhaNER with varying translation data sizes.

| Translation Data Size | Trans-train | GoLLIE-TF |
|---|---|---|
| 1,000 | 52.8 | **62.4** |
| 5,000 | 52.6 | **61.2** |
| 10,000 | 54.9 | **62.7** |
| 40,000 | 56.4 | **66.3** |

**Robustness to translation quality.** TransFusion offers a distinct advantage by leveraging an external multilingual MT system to augment its dataset with English translations. However, the efficacy of this approach hinges on the translation quality provided by the external MT system.

In Figure 3, we explore this aspect by evaluating GoLLIE-TF's performance with three different MT systems (NLLB-200-600m, 1.3b, 3.3b) and use Flores-200 translation benchmark (X to English) (Costa-jussà et al., 2022) to measure translation quality (spBLUE) of languages covered by MasakhaNER2. Our experiments reveal that GoLLIE-TF exhibits robustness across various MT systems, as we observe that the F1 score on MasakhaNER2 does not exhibit a significant drop, however performance does improve with a stronger translation system.

## 5.2 ENHANCING GPT-4 WITH TRANSFUSION

Despite GPT-4's pre-training on multilingual corpora, a notable performance gap persists between its English NER capabilities on CoNLL03 (80 F1) and its performance on low-resource languages (54.2 F1). In Figure 4, we employ the TransFusion instruction, asking GPT-4 for predictions on the English translation and to then use these labels to predict on the target language sentence. We show TransFusion prompting yields a substantial enhancement in GPT-4's NER performance across MasakhaNER2 and three additional low-resource languages from the UNER dataset (Cebuano, Tagalog-Philippines, and Uganda), improving the average F1 score from 53.4 to 62. This shows the GPT-4 can follow TransFusion prompting framework to leverage its English predictions to make more accurate predictions on low-resource languages.

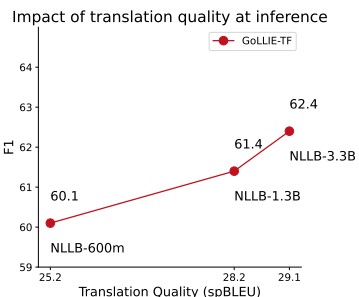

Figure 3: TransFusion robustness to different translation systems.

## 5.3 TRANSFUSION WITH ENCODER-ONLY MODELS

We have demonstrated that TransFusion can be applied to GPT-4 to improve low-resource language NER performance and also with the decoder-only LLM GoLLIE, which has the benefit of generalizing to unseen label schemas. In this section, we experiment with encoder-only multilingual LMs (Devlin, 2018) as the encoder architecture is one of the standard approaches for NER tasks used in practice.

As encoder-only models generally assume the same label schema between fine-tuning and evaluation, we focus on the seen label schema experiment setting, where we use CoNLL03 English as training data and test on the full test set of MasakhaNER2. We use AfroXLM-R (Alabi et al., 2022), an African language pre-trained language model as MasakhaNER is an African language dataset. For each language, we fine-tuned the model on a combination (50/50%) of English and translation (Trans-train) or TransFusion data for 5 epochs with a learning rate of 2e-5. The specific TransFusion implementation is introduced in Section 3.1.

In Table 5, we show the effectiveness of the TransFusion framework which boosts the F1 from 58.8 to 72.1 F1 on MasakhaNER2 with AfroXLM-R. In addition, it outperforms the Trans-train baseline significantly with a +6.3 F1 improvement and achieves state-of-the-art performance on MasakhaNER2,

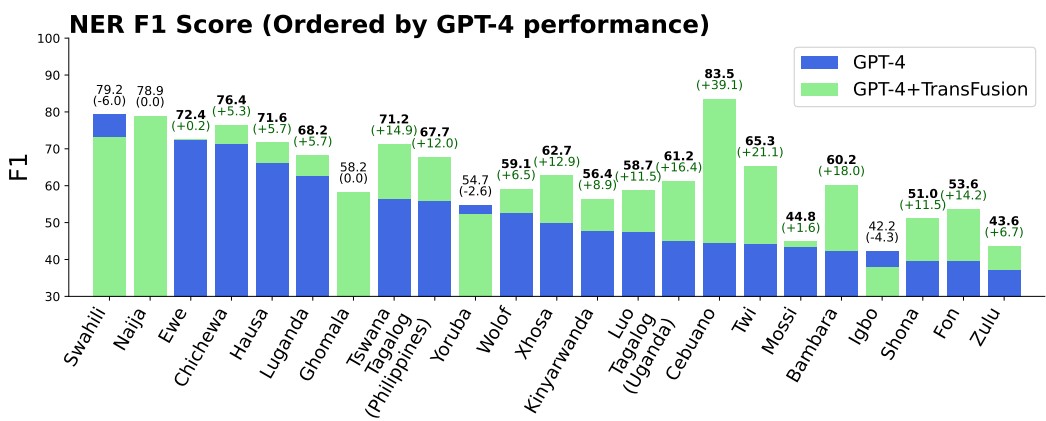

Figure 4: GPT-4 + TransFusion framework improves NER on low-resource language from MasakhaNER2 and UNER subsets. On average, GPT-4 + TransFusion improves average F1 from 53.4 to 62.

Table 5: F1 of encoder-only multilingual LM on MasakhaNER2, average of 3 random seeds. Avg (CLaP) shows the average of F1 over nine languages reported in CLaP.

| Model | Avg (CLaP) | Avg (all) |
|---|---|---|
| **Translate-train** | | |
| EasyProject (Chen et al., 2023b) | 67.2 | 64.9 |
| CLaP (Parekh et al., 2023) | 58.8 | - |
| **Translate-test** | | |
| Awesome-align (Dou & Neubig, 2021) | 67.0 | 65.8 |
| CoDec (Le et al., 2024) | 73.9 | 70.4 |
| **TransFusion** (ours) | **74.2** | **72.0** |

surpassing the previous state-of-the-art Codec (Le et al., 2024). Codec uses constrained decoding within a translation model to generate precise label projections from English to the target language for Translate-test. In contrast, TransFusion introduces a model that learns to fuse annotations, showing robustness to errors in English annotation predictions. Overall, this shows the generalization of the TransFusion to the encoder-only multilingual language model.

## 5.4 Error Analysis

To understand the reasons why GoLLIE-TF makes mistakes, we conducted a manual error analysis on the MasakhaNER2 (Akan) subset and annotated 31 errors from the model. In Figure 5, we show examples of two common error types made by GoLLIE-TF: (1) English prediction errors, where the predictions on English translation are incorrect, and (2) Fusion errors, where the error arises from the fusion stage. We identified 22 out of 31 cases where the model made errors in predicting NER for the English translation text, and thus these errors propagated to the final predictions. On the other hand, we found 12 out of 31 cases where the model made incorrect fusion processes, leading to hallucinations in the final predictions or predictions in the English text.

## 6 Related Work

**Multilingual language models.** Multilingual language models (Devlin, 2018; Conneau & Lample, 2019; Conneau et al., 2020; Xue et al., 2021; Scao et al., 2022; Asai et al., 2023), have facilitated cross-lingual transfer by leveraging pre-training on large-scale multilingual corpora. Recent models such as Gemini (Gemini et al., 2023) show emergent capabilities such as ultra low-resource language translation with a book and wordlist in context. However, their performance tends to be subpar on languages that were not seen during pre-training or are underrepresented in the training data (Adelani et al., 2021; Ebrahimi et al., 2022). To address this limitation, several approaches have been explored, including bilingual models (Lan et al., 2020; Wang et al., 2020), language-specific extensions (Ogueji

| Error Type | Target Text | English Translation | Gold | English Prediction | Final Prediction |
|---|---|---|---|---|---|
| English Prediction Error | Mehyɛ mo nyinaa bɔ sɛ yei yɛ nneɛma akɛsea mfitiaseɛ ma Ghana Mmaranim Sukuu no . Aban bɔhyɛ sɛ ɔbɛgya biribi ama nkyirmma wɔ' | I promise you all that this is a great beginning for the Ghana School of Law | **LOC**: Ghana | **ORG**: Ghana School of Law | **ORG**: Ghana Mmaranim Sukuu no |
| English Prediction Error | ka kyerɛɛ asɛnnibea sɛ Yeboah de nkuu bi ɛhyehye faa abɔfra no ayaase de ne nsa wowɔɔ nase ansa ɔreto no mmonaa | Ntee said to the court that Yeboah took a burning torch to the child's throat and rubbed his nose with his hand before kissing him | **PER**: Yeboah | **PER**: Ntee
**PER**: Yeboah | *Error Propagation*
**PER**: Ntee
**PER**: Yeboah |
| English Prediction + Fusion Error | Mɛka akyerɛ Ghana manfoɔ nyinaa ara sɛ yɛretu anamɔn a ɛho hia biara sɛ yɛbɛhwɛ ama nnipakan dwumadie yi bɛdi COVID - 19 banbɔ nhyehyɛeɛ so . Nneɛma bɛn na yɛreyɛ ? Yadikan ne Ghana Apɔmuden Asoeɛ anya nkitahodie na wɔn ne Dr . Annthony Nsiah Asare a ɔyɛ' | I would like to inform all Ghanaians that we are taking all necessary steps to ensure that this census is conducted in accordance with the COVID - 19 safety protocols. What steps are we taking? Yadikan has been in contact with | **LOC**: Ghana
**PER**: Anthony Nsiah Asare
**ORG**: Apɔmuden Asoeɛ | **ORG**: Yadikan
**PER**: Annthony Nsiah Asare
**ORG**: Ministry of Health | *Error Propagation*
**ORG**: Yadikan
**PER**: Annthony Nsiah Asare
**ORG**: Ministry of Health *Prediction in English* |
| Fusion Error | Sɛ́ Asamoah da so ara wɔ ɔsram biako bio a ɛsɛ sɛ ɔko ansa na wawie sukuu | Asamoah still has one more month to go before he graduates | **PER**: Asamoah | **PER**: Asamoah | **PER**: Sɛ́ Asamoah da so ara wɔ ɔsram… *Hallucination* |

Figure 5: **Error analysis** of GoLLIE-TF's 31 incorrect predictions on MasakhaNER2 (Akan). Two common errors are categorized as English prediction error (22/31) and fusion error (12/31).

et al., 2021; Alabi et al., 2022; Yoon et al., 2024), continued training (Wang et al., 2020; Pfeiffer et al., 2020; Wang et al., 2022; Imani et al., 2023), and few-shot learning (Lin et al., 2022). Recently, multilingual instruction-tuning (Chen et al., 2023a) datasets such as Aya (Singh et al., 2024; Üstün et al., 2024) focusing on text generation and IEPile (Gui et al., 2024) (English and Chinese) have been proposed to facilitate this direction of research.

**Translation for cross-lingual transfer.** To enhance LLM on multilingual NLP tasks such as QA (Agrawal et al., 2023), translating train or test data (Artetxe et al., 2023) into English has proven as an effective approach (Paolini et al., 2021; Hu et al., 2020; Xue et al., 2021; Ebing & Glavaš, 2024; Ansell et al., 2023). Recent studies on prompting LLMs with translation demonstrate improvements on multilingual math reasoning (Shi et al., 2022), text generation (Huang et al., 2023; Intrator et al., 2024; Liu et al., 2024) and sentence classification (Etxaniz et al., 2023). In contrast, our work focuses on challenging IE tasks that require extracting span annotations on the target language directly, instead of generating text. It is even more challenging to construct translated data for translate-train as span annotations are missing after translation. To solve this, word alignment models (Och & Ney, 2003; Dyer et al., 2013; Lan et al., 2021; Dou & Neubig, 2021; Parekh et al., 2023; Le et al., 2024) and a simple mark-then-translate approach (Lee et al., 2018; Lewis et al., 2020; Hu et al., 2020; Bornea et al., 2021; Chen et al., 2023b) have been utilized to project labels across different languages. In contrast, we train a model to fuse annotations from English and directly make predictions on target language.

## 7 CONCLUSION

We introduce TransFusion, a framework that bridges the performance gap between high and low-resource languages in information extraction by leveraging machine translation. Our experiments demonstrate that TransFusion significantly improves the cross-lingual transfer capabilities of instruction-tuned LLMs, surpassing both proprietary models and encoder-only architectures on low-resource languages NER. This work demonstrates the potential of translation-based techniques to unlock the power of LLMs for a wider range of low-resource languages, paving the way for more inclusive and equitable IE capabilities across diverse linguistic communities.

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

## A APPENDIX

Table 6: Evaluation datasets used and the language code for each dataset.

| Dataset | Language Code |
|---|---|
| MasakhaNER2.0 (Adelani et al., 2022)

afl-3.0 License

`masakhane/masakhaner2` | Bambara (bam), Ghomala (bbj), Ewe (ewe), Fon (fon), Hausa (hau),
Igbo (ibo), Kinyarwanda (kin), Luganda (lug), Luo (luo), Mossi (mos),
Nyanja (nya), Naija (pcm), Shona (sna), Swahili (swh), Tswana (tsn)
Twi (twi), Wolof (wol), Xhosa (xho), Yoruba (yor), Zulu (zul) |
| UNER (Mayhew et al., 2023)
`universalner.org/`
(Unknown License) | Cebuano (ceb_gja), Danish (da_ddt), German (de_pud), English (en_ewt), English (en_pud), Croatian (hr_set), Portuguese (pt_bosque), Portuguese (pt_pud), Russian (ru_pud), Slovak (sk_snk), Serbian (sr_set), Swedish (sv_pud), Swedish (sv_talbanken), Tagalog (tl_trg), Tagalog (tl_ugnayan), Chinese (zh_gsd), Chinese (zh_gsdsimp), Chinese (zh_pud) |
| ACE05 (Walker et al., 2006)
LDC license: LDC2006T06 | English (en), Arabic (ar), Chinese (zh) |
| MultiNERD (Tedeschi & Navigli, 2022)
CC BY-NC-SA 4.0
`Babelscape/multinerd` | German (de), Spanish (es), French (fr), Italian (it), Dutch (nl), Polish (pl), Portuguese (pt), Russian (ru), Chinese (zh) |
| MultiCoNER2 (Fetahu et al., 2023)

CC BY 4.0
`MultiCoNER/multiconer_v2` | Bengali (bn), German (de), Spanish (es), Persian (fa), French (fr), Hindi (hi), Italian (it), Portuguese (pt), Swedish (sv), Ukrainian (uk), Chinese (zh), English (en) |
| xSID (van der Goot et al., 2021)

CC BY-SA 4.0 | Arabic (ar), Danish (da), German (de), English (en), Indonesian (id), Italian (it), Japanese (ja), Kazakh (kk), Dutch (nl), Serbian (sr), Turkish (tr), Chinese (zh) |
| MultiTO (Schuster et al., 2018)
CC-BY-SA | English (en), Spanish (es), Thai (th) |
| RED-FM (Cabot et al., 2023)
CC BY-SA 4.0
`Babelscape/REDFM` | Arabic (ar), German (de), English (en), Spanish (es), French (fr), Italian (it), Chinese (zh) |
| MASSIVE (FitzGerald et al., 2022)

CC BY 4.0

`AmazonScience/massive` | Afrikaans (af-za), Amharic (am-et), Azeri (az-za), Bengali (bn-bd), Armenian (hy-am), Georgian (ka-ge),Khmer (km-kh), Mongolian (mn-mn), Burmese (my-mm), Kannada (kn-in), Malayalam (ml-in), Tamil (ta-in), Telugu (te-in), Tagalog (tl-ph), Welsh (cy-gb) |

## B LIMITATIONS AND BROADER IMPACT

The NER experiments conducted on GPT-4 have yielded promising results for low-resource languages. However, concerns remain regarding potential data contamination resulting from the possibility that GPT-4 was pre-trained or fine-tuned on the test data.[1] The Translation-and-fusion framework, while effective in enhancing cross-lingual transfer, does introduce additional inference costs during test time inference. These additional steps include translation using an external MT system and annotation processes, which can contribute to an increased number of token generation. This is similar to chain-of-thought prompting or retrieval augmented generation, which uses additional computational cost at inference for better quality generation. Thus, practitioners should consider the trade-off between performance and efficiency when deciding to adopt the Translation-and-fusion approach. We show an estimate of inference time costs in Table 7.

---

[1]https://hitz-zentroa.github.io/lm-contamination/blog/

Figure 6: Example of input and output representation. (left) An example of a named entity recognition prompt and output annotations. (right) The same example but with translation text appended in the input prompt with instructions to guide the model to generate annotations on English translation text first, followed by annotations on the target language.

Figure 7: Examples of GoLLIE-TF model generation out (colored in gray).

The proposed method carries minimal risk, given that it addresses a traditional IE task. Its primary objective is to enhance IE cross-lingual transfer performance for low-resource languages lacking annotated training data. Consequently, our work aims to have a broader impact by facilitating research for global communities with diverse languages.

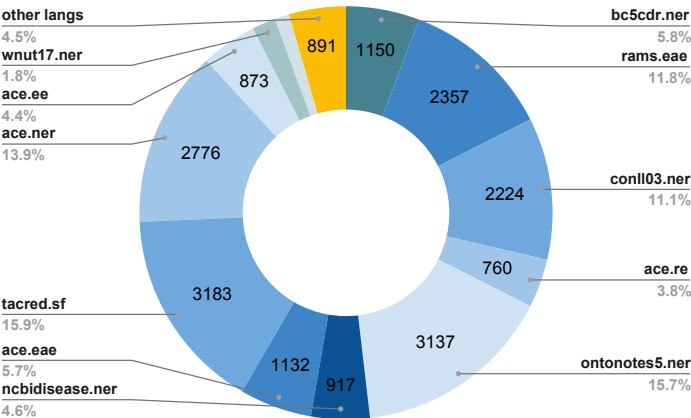

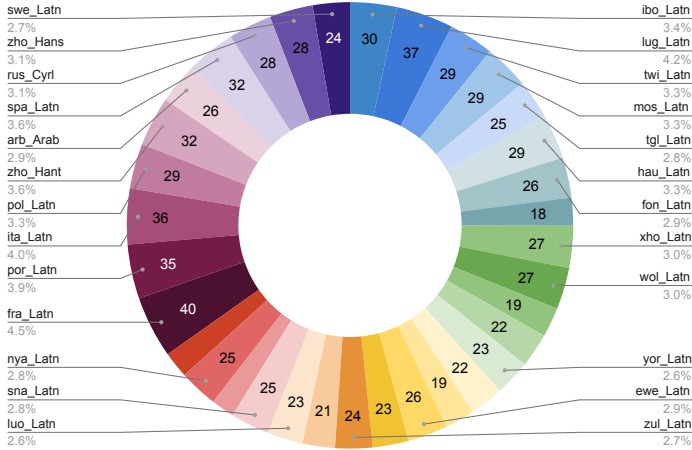

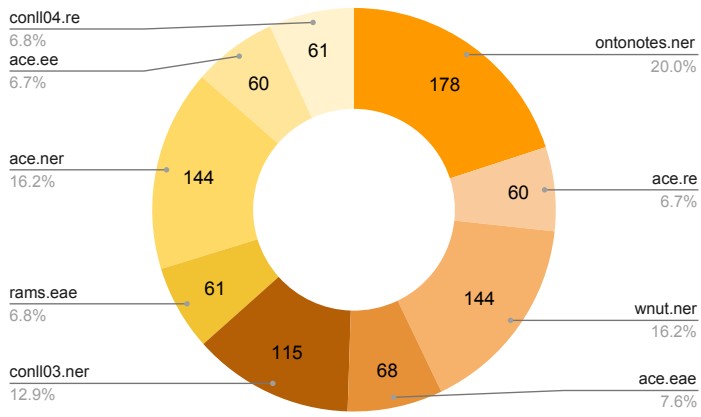

Figure 8: TransFusion training dataset mixture for a total of 20,000.

Table 7: Inference time (seconds/sentence) cost comparison of GoLLIE and GoLLIE-TF models on a single NVIDIA A40 GPU.

| Dataset | Language | Model | F1 Score | Inference Time | MT Time | Total Time |
|---|---|---|---|---|---|---|
| MasakhaNER | Bambara | GoLLIE | 38.9 | 0.58 | 0 | 0.58 |
| MasakhaNER | Bambara | GoLLIE-TF | 54.8 | 1.11 | 0.285 | 1.395 |
| Massive | Bengali | GoLLIE | 5.7 | 0.555 | 0 | 0.555 |
| Massive | Bengali | GoLLIE-TF | 18.1 | 0.705 | 0.08 | 0.785 |

Table 8: We report GoLLIE-TF on MasakhaNER2 and Massive for 3 different seeds.

| Dataset | Seed 0 | Seed 1 | Seed 2 | Mean | Std dev |
|---|---|---|---|---|---|
| masakhaner.bam.ner | 54.8 | 53.7 | 56.1 | 54.9 | 1.2 |
| masakhaner.bbj.ner | 50.2 | 46.2 | 50.9 | 49.1 | 2.6 |
| masakhaner.ewe.ner | 73.2 | 72.7 | 73.1 | 73.0 | 0.3 |
| masakhaner.fon.ner | 57.9 | 54.3 | 55.7 | 56.0 | 1.8 |
| masakhaner.hau.ner | 67.1 | 65.6 | 66.2 | 66.3 | 0.8 |
| masakhaner.ibo.ner | 56.6 | 54.2 | 55.7 | 55.5 | 1.3 |
| masakhaner.kin.ner | 58.5 | 59.5 | 59.6 | 59.2 | 0.6 |
| masakhaner.lug.ner | 75.5 | 74.5 | 75.1 | 75.0 | 0.5 |
| masakhaner.luo.ner | 51.7 | 51.6 | 51.5 | 51.6 | 0.1 |
| masakhaner.mos.ner | 48.8 | 43.8 | 44.4 | 45.7 | 2.7 |
| masakhaner.nya.ner | 78.2 | 78.7 | 78.9 | 78.6 | 0.3 |
| masakhaner.pcm.ner | 81.1 | 80.8 | 80.6 | 80.8 | 0.2 |
| masakhaner.sna.ner | 57.4 | 59.2 | 56.7 | 57.7 | 1.3 |
| masakhaner.swh.ner | 73.5 | 72.6 | 72.9 | 73.0 | 0.5 |
| masakhaner.tsn.ner | 71.0 | 70.3 | 71.1 | 70.8 | 0.5 |
| masakhaner.twi.ner | 74.2 | 68.6 | 76.6 | 73.1 | 4.1 |
| masakhaner.wol.ner | 61.9 | 55.6 | 60.2 | 59.2 | 3.2 |
| masakhaner.xho.ner | 49.9 | 54.4 | 51.3 | 51.9 | 2.3 |
| masakhaner.yor.ner | 54.4 | 52.4 | 53.4 | 53.4 | 1.0 |
| masakhaner.zul.ner | 52.8 | 53.3 | 51.4 | 52.5 | 1.0 |
| Average | 62.4 | 61.1 | 62.1 | 61.9 | 0.7 |
| massive.en-us.ner | 53.6 | 51.6 | 51.6 | 52.3 | 1.1 |
| massive.af-za.ner | 24.2 | 21.2 | 24.2 | 23.2 | 1.7 |
| massive.am-et.ner | 6.5 | 5.4 | 7.2 | 6.4 | 0.9 |
| massive.az-az.ner | 1.2 | 1.3 | 1.3 | 1.2 | 0.1 |
| massive.bn-bd.ner | 18.1 | 18.8 | 19.4 | 18.8 | 0.6 |
| massive.hy-am.ner | 19.4 | 16.2 | 21.1 | 18.9 | 2.5 |
| massive.ka-ge.ner | 18.4 | 16.0 | 19.6 | 18.0 | 1.9 |
| massive.km-kh.ner | 20.4 | 21.1 | 23.2 | 21.5 | 1.5 |
| massive.mn-mn.ner | 5.8 | 5.4 | 5.2 | 5.5 | 0.3 |
| massive.my-mm.ner | 31.7 | 32.4 | 33.2 | 32.4 | 0.8 |
| massive.kn-in.ner | 17.2 | 14.2 | 20.7 | 17.3 | 3.2 |
| massive.ml-in.ner | 11.0 | 10.6 | 10.3 | 10.7 | 0.4 |
| massive.ta-in.ner | 17.0 | 11.6 | 17.3 | 15.3 | 3.2 |
| massive.te-in.ner | 18.8 | 17.6 | 23.5 | 20.0 | 3.1 |
| massive.tl-ph.ner | 32.0 | 32.0 | 34.7 | 32.9 | 1.5 |
| massive.cy-gb.ner | 8.3 | 5.8 | 7.0 | 7.0 | 1.2 |
| Average | 19.0 | 17.6 | 20.0 | 18.8 | 1.2 |

Table 9: Full experimental results (1) for each dataset and language. Format: [task name].[language code].[task].

|  | GPT-4 | GoLLIE | Trans-train | GoLLIE-TF (ours) |
|---|---|---|---|---|
| uner.ceb_gja.ner | 44.4 | 49.6 | 52.9 | 87.5 |
| uner.da_ddt.ner | 77.2 | 76.7 | 79.4 | 84.8 |
| uner.de_pud.ner | 80.3 | 80.1 | 82.3 | 83.8 |
| uner.en_ewt.ner | 59.9 | 84.7 | 67.6 | 66.4 |
| uner.en_pud.ner | 75.4 | 82.4 | 85.5 | 84.9 |
| uner.hr_set.ner | 82.1 | 83.0 | 87.7 | 89.6 |
| uner.pt_bosque.ner | 82.7 | 84.5 | 84.2 | 81.3 |
| uner.pt_pud.ner | 80.5 | 87.2 | 89.6 | 90.3 |
| uner.ru_pud.ner | 69.8 | 68.3 | 71.6 | 73.3 |
| uner.sk_snk.ner | 70.9 | 71.2 | 81.4 | 85.5 |
| uner.sr_set.ner | 85.9 | 86.2 | 88.5 | 88.9 |
| uner.sv_pud.ner | 73.7 | 81.5 | 79.6 | 85.7 |
| uner.sv_talbanken.ner | 68.7 | 69.4 | 64.6 | 75.7 |
| uner.tl_trg.ner | 55.7 | 58.8 | 60.3 | 54.2 |
| uner.tl_ugnayan.ner | 44.8 | 61.0 | 57.1 | 74.2 |
| uner.zh_gsd.ner | 60.6 | 62.5 | 58.8 | 67.6 |
| uner.zh_gsdsimp.ner | 57.9 | 62.4 | 61.4 | 68.8 |
| uner.zh_pud.ner | 72.0 | 74.8 | 72.6 | 77.7 |
| average | 69.0 | 73.6 | 73.6 | 78.9 |
| ace.en.eae | 24.5 | 97.3 | 97.9 | 98.3 |
| multiace.ar.eae | 1.6 | 84.3 | 83.8 | 81.8 |
| multiace.zh.eae | 9.6 | 96.6 | 97.1 | 77.9 |
| average | 11.7 | 92.7 | 92.9 | 86.0 |
| ace.en.ee | 27.8 | 67.5 | 64.0 | 60.4 |
| multiace.ar.ee | 24.4 | 16.1 | 12.8 | 25.0 |
| multiace.zh.ee | 11.6 | 44.2 | 43.3 | 46.7 |
| average | 21.3 | 42.6 | 40.0 | 44.0 |
| ace.en.ner | 58.0 | 78.3 | 87.3 | 86.5 |
| multiace.ar.ner | 32.3 | 29.5 | 30.3 | 37.5 |
| multiace.zh.ner | 34.6 | 68.2 | 66.0 | 60.6 |
| average | 41.6 | 58.7 | 61.2 | 61.5 |
| ace.en.re | 5.40 | 58.2 | 59.8 | 58.1 |
| multiace.ar.re | 3.2 | 14.1 | 13.5 | 15.8 |
| multiace.zh.re | 5.1 | 39.5 | 44.8 | 43.3 |
| average | 4.6 | 37.3 | 39.4 | 39.1 |
| multinerd.de.ner | 75.8 | 69.3 | 73.2 | 74.4 |
| multinerd.es.ner | 69.4 | 72.0 | 68.1 | 69.5 |
| multinerd.fr.ner | 71.8 | 71.9 | 74.4 | 72.5 |
| multinerd.it.ner | 76.2 | 69.8 | 74.2 | 70.5 |
| multinerd.nl.ner | 76.9 | 67.8 | 73.0 | 72.5 |
| multinerd.pl.ner | 72.1 | 62.0 | 64.0 | 61.5 |
| multinerd.pt.ner | 67.7 | 67.7 | 66.3 | 64.9 |
| multinerd.ru.ner | 65.3 | 57.9 | 55.7 | 58.7 |
| multinerd.zh.ner | 7.8 | 7.1 | 13.9 | 8.8 |
| multinerd.ner | 71.5 | 76.2 | 75.6 | 76.2 |
| average | 71.9 | 62.2 | 63.9 | 63.0 |

Table 10: Full experimental results (2) for each dataset and language. Format: [task name].[language code].[task].

| | GPT-4 | GoLLIE | Trans-train | GoLLIE-TF (ours) |
|---|---|---|---|---|
| multiconer2.bn.ner | 43.9 | 2.7 | 7.9 | 27.6 |
| multiconer2.de.ner | 54.4 | 27.3 | 30.8 | 33.1 |
| multiconer2.es.ner | 44.8 | 18.1 | 23.9 | 26.1 |
| multiconer2.fa.ner | 40.1 | 15.6 | 34.9 | 41.4 |
| multiconer2.fr.ner | 54.2 | 29.2 | 32.1 | 34.2 |
| multiconer2.hi.ner | 46.9 | 5.0 | 14.8 | 33.5 |
| multiconer2.it.ner | 51.1 | 41.4 | 46.0 | 46.5 |
| multiconer2.pt.ner | 49.7 | 23.6 | 31.5 | 34.7 |
| multiconer2.sv.ner | 52.5 | 14.8 | 16.1 | 19.6 |
| multiconer2.uk.ner | 55.9 | 41.1 | 47.7 | 51.7 |
| multiconer2.zh.ner | 5.1 | 14.0 | 20.9 | 28.3 |
| multiconer2.en.ner | 54.6 | 34.1 | 34.7 | 36.7 |
| average | 46.1 | 22.2 | 28.4 | 34.5 |
| xsid.ar.ner | 53.2 | 0.0 | 29.7 | 28.7 |
| xsid.da.ner | 48.1 | 2.7 | 15.5 | 16.0 |
| xsid.de.ner | 48.9 | 9.8 | 36.0 | 35.5 |
| xsid.en.ner | 63.1 | 28.8 | 38.4 | 37.5 |
| xsid.id.ner | 49.4 | 0.7 | 25.6 | 23.2 |
| xsid.it.ner | 52.1 | 3.4 | 30.2 | 32.8 |
| xsid.ja.ner | 28.1 | 10.1 | 32.8 | 26.5 |
| xsid.kk.ner | 34.9 | 0.0 | 0.0 | 2.5 |
| xsid.nl.ner | 48.9 | 4.9 | 33.8 | 31.4 |
| xsid.sr.ner | 48.7 | 0.0 | 19.4 | 16.8 |
| xsid.tr.ner | 40.8 | 0.8 | 20.9 | 22.2 |
| xsid.zh.ner | 47.3 | 10.7 | 43.5 | 43.7 |
| average | 47.0 | 6.0 | 27.1 | 26.4 |
| multito.en.ner | 51.1 | 35.3 | 39.0 | 40.3 |
| multito.es.ner | 1.4 | 2.5 | 3.0 | 2.3 |
| multito.th.ner | 7.3 | 15.4 | 18.9 | 11.8 |
| average | 19.9 | 17.7 | 20.3 | 18.1 |
| redfm.ar.re | 18.3 | 11.6 | 9.0 | 13.9 |
| redfm.de.re | 31.0 | 22.3 | 24.8 | 13.1 |
| redfm.en.re | 19.9 | 14.8 | 18.6 | 15.7 |
| redfm.es.re | 17.4 | 13.8 | 18.6 | 14.4 |
| redfm.fr.re | 17.1 | 15.2 | 19.2 | 17.6 |
| redfm.it.re | 17.2 | 20.0 | 17.1 | 29.1 |
| redfm.zh.re | 12.9 | 10.4 | 10.5 | 9.7 |
| average | 19.1 | 15.5 | 16.8 | 16.2 |

Table 11: Full experimental results (3) for each dataset and language. Format: [task name].[language code].[task].

|  | GPT-4 | GoLLIE | Trans-train | GoLLIE-TF (ours) |
|---|---|---|---|---|
| massive.en-us.ner | 55.2 | 45.9 | 54.7 | 53.6 |
| massive.af-za.ner | 52.6 | 8.2 | 23.4 | 24.2 |
| massive.am-et.ner | 17.0 | 0.0 | 0.8 | 6.5 |
| massive.az-az.ner | 25.7 | 4.0 | 11.0 | 1.2 |
| massive.bn-bd.ner | 33.1 | 5.7 | 13.0 | 18.1 |
| massive.hy-am.ner | 33.6 | 1.2 | 11.9 | 19.4 |
| massive.ka-ge.ner | 32.1 | 10.4 | 12.2 | 18.4 |
| massive.km-kh.ner | 33.9 | 0.0 | 11.3 | 20.4 |
| massive.mn-mn.ner | 19.5 | 0.0 | 5.3 | 5.8 |
| massive.my-mm.ner | 27.9 | 4.8 | 15.2 | 31.7 |
| massive.kn-in.ner | 33.1 | 0.0 | 2.6 | 17.2 |
| massive.ml-in.ner | 25.1 | 0.0 | 4.5 | 11.0 |
| massive.ta-in.ner | 30.7 | 1.2 | 5.0 | 17.0 |
| massive.te-in.ner | 28.7 | 0.0 | 0.0 | 18.8 |
| massive.tl-ph.ner | 50.3 | 12.3 | 20.2 | 32.0 |
| massive.cy-gb.ner | 33.6 | 0.0 | 3.1 | 8.3 |
| average | 33.3 | 5.9 | 12.1 | 19.0 |

Table 12: Comparison of GPT-4 and GPT-4+Transfusion.

| Language | GPT-4 | GPT-4+Transfusion |
|---|---|---|
| MasakhaNER2 |  |  |
| bam | 42.2 | **60.2** |
| bbj | **58.2** | 52.9 |
| ewe | 72.2 | **72.4** |
| fon | 39.4 | **53.6** |
| hau | 65.9 | **71.6** |
| ibo | **42.2** | 37.9 |
| kin | 47.5 | **56.4** |
| lug | 62.5 | **68.2** |
| luo | 47.2 | **58.7** |
| mos | 43.2 | **44.8** |
| nya | 71.1 | **76.4** |
| pcm | **78.9** | 75.7 |
| sna | 39.5 | **51.0** |
| swh | **79.2** | 73.2 |
| tsn | 56.3 | **71.2** |
| twi | 44.2 | **65.3** |
| wol | 52.6 | **59.1** |
| xho | 49.8 | **62.7** |
| yor | **54.7** | 52.1 |
| zul | 36.9 | **43.6** |
| MasakhaNER2 average | 54.2 | **59.9** |
| UNER |  |  |
| ceb_gja | 44.4 | **83.5** |
| tl_trg | 55.7 | **67.7** |
| tl_ugnayan | 44.8 | **61.2** |
| All average | 53.4 | **62.0** |

