# OpenReview forum: "Translation and Fusion Improves Zero-shot Cross-lingual Information Extraction"
_ICLR.cc/2025/Conference — Submitted to ICLR 2025_

### Official Review · Reviewer_T2WB · 2024-10-22

**Soundness:** 3
**Presentation:** 3
**Contribution:** 2
**Rating:** 5
**Confidence:** 4

**Summary:**

- The study proposes a translation and fusion (TransFusion) method as an extension to Gollie for zero-shot cross-lingual information extraction.
- The method involves appending translated target language instances to the input prompt. The model is prompted to perform information extraction in English first, which serves as a reference label to improve information extraction performance in the target language.
- For encoder-only language models this entails a two-step approach, where a model fine-tuned in English is used to provide the reference labels for the translated text first. The translated text with the reference labels is then concatenated with the inputs in the target languages and used for the final prediction outcome of the target language data. For decoder-only models, the two steps are combined in one prompt by providing instructions to first generate reference labels in English and then perform the task in the target language.

**Strengths:**

- The evaluation results suggest performance gains for low-resource languages in comparison to GPT-4, regular Gollie and translate-train baselines.
- The performance benchmarks includes various information extraction tasks across 12 datasets and low-resource languages. Specifically the performance benchmark on low-resource languages shows promising performance benefits for prompting decoder-only and fine-tuning encoder-only models.
- The authors provide ablation studies evaluating the contribution of the annotation in English and the impact of different sizes of machine translation models.

**Weaknesses:**

- The benchmarks lack the translate-test baseline. The annotations of the translated English text are not formally included in the evaluation. This evaluation would provide insights in the contribution of the “fusion” step of the framework and should be included for both GPT-4 and Gollie.
- The proposed method lacks innovation. Providing annotated translations in the input prompt to enhance multilingual performance has been explored in related work [1,2]. Furthermore, the engineering approach to cross-lingual information extraction lacks detailed evaluation of each individual component.
- The discussion of the additional computational complexity introduced through machine translation and annotation process is not included in the main body of the paper (included in Appendix Table 7).

References

- [1] Xi Victoria Lin, Todor Mihaylov, Mikel Artetxe, Tianlu Wang, Shuohui Chen, Daniel Simig, Myle Ott, Naman Goyal, Shruti Bhosale, Jingfei Du, Ramakanth Pasunuru, Sam Shleifer, Punit Singh Koura, Vishrav Chaudhary, Brian O’Horo, Jeff Wang, Luke Zettlemoyer, Zornitsa Kozareva, Mona Diab, et al.. 2022. Few-shot Learning with Multilingual Generative Language Models. In Proceedings of the 2022 Conference on Empirical Methods in Natural Language Processing, pages 9019–9052, Abu Dhabi, United Arab Emirates. Association for Computational Linguistics.
- [2] Priyanka Agrawal, Chris Alberti, Fantine Huot, Joshua Maynez, Ji Ma, Sebastian Ruder, Kuzman Ganchev, Dipanjan Das, and Mirella Lapata. 2023. QAmeleon: Multilingual QA with Only 5 Examples. Transactions of the Association for Computational Linguistics, 11:1754–1771.

**Questions:**

- Did you consider skipping the translation step during inference for the fine-tuned Gollie-TF? It would be interesting whether training on translation + fusion already results in performance improvements without the need to translate and annotate the test data.
- In lines 48 you mention that your TransFusion method trains language models on tools usage (MT model), however your method does not integrate an interactive use of the MT model. Can you elaborate this description?

---

> ### Author Response · Authors · 2024-11-25
>
> - The Translate-test baseline is included in Table 5 using awesome-align and CoDec. For the GPT-4 and GoLLIE experiments, we did not implement Translate-test with word alignment because it requires significant manual effort to design heuristic rules for the label projection stage for each dataset and task.
>
> - Thank you for providing two related works [1] [2] and we have cited them in the related work (L513, L518). Our work primarily focuses on information extraction (IE) tasks, which require predictions at the word-span level. We provide evaluations using different MT systems, varying amounts of training data, and three different configurations (LLM, LM, and prompting), as shown in Table 3 (w/o annotation during inference).
>
> - We are open to discussing the inference time cost and have specifically provided data on this topic in the appendix due to space limitations (Page 16, Section B).
>
> - Question 1: We experimented with this setting in Table 3 (no translation) and it shows the same performance as the GoLLIE model. The model follows the original GoLLIE instruction and predicts the annotations directly. This shows the benefits of using translation.
>
> - L48: The phrase "teach models to use an external translation model to make better predictions" can be rephrased as: "teach models to use translation data from an external MT system to make better predictions." (L048).

---

> > ### Comment · Reviewer_T2WB · 2024-11-26
> > **Response to Authors**
> >
> > Thank you for the clarifications and additional ablation experiments. I adjusted my evaluation accordingly.

---

### Official Review · Reviewer_ow8F · 2024-11-04

**Soundness:** 3
**Presentation:** 3
**Contribution:** 2
**Rating:** 6
**Confidence:** 4

**Summary:**

This work introduces a new framework to deal with information extraction in the context of zero shot cross-lingual approaches on low resource regimes. The framework, called TransFusion, consists in two parts i) rely on a high resource language model that that is fed with the translation of the source part in the original language and ii) to fuse the response in the high resource language and the portion on the low resource one training a model to predict an answer based on both inputs.
The framework is applied to the existing English-based model, GoLLIE, to produce GoLLIE-TF, a model to deal with cross-lingual information extraction for low resource languages.
The resulting model is run on a variety of datasets, covering tasks such as NER, Slot detection and relation extraction on up to 50 different languages.

**Strengths:**

The approach is effective and can be applied in a variety of configurations, i.e in decoder only models, on a large pretrained model like GPT-4 to be used zero shot on even on an encoder only model.
The work makes a good effort in covering several low resource languages from available datasets.
The approach improves over GPT-4 and for translate-traine baselines in most of the languages.

**Weaknesses:**

Although the approach is relatively simple and straightforward it is not clearly explained on the paper and requires a thorough reading to understand it (see below).
One of the salient points of this paper could also be considered a drawback. TransFusion has been shown to be flexible enough to be used in three different scenarios, each one of them having some minor differences in implementation. One could argue these three "flavours"of TransFusion are, in fact, different models (see below)

**Questions:**

* Figure 1 could have shown more clearly the fact that the initial portion of the text is translated into English, ran on an english-based model and then with its output and the portion in the original language, the output is predicted. None of these aspects (nor the fact the TransFusion module is trained separately) is shown in this image.

* Please fix the reference "Team et all, 2023", which took the "Gemini Team" as the name of the first author, and "Team" as its last name

* Although it's not an issue, a better presentation would have completely filled the 10 pages in the submission.

* line: 419  "lagnauge"

* line 423 fix double parenthesis. Authors might be referring about two varieties of Tagalog, from Philippines and from Uganda

* Authors might like to stress the fact that the different configurations of TransFusion are, indeed, different usages of the same approach.

**Details Of Ethics Concerns:**

--

---

> ### Author Response · Authors · 2024-11-25
>
> - In Figure 1, we show both input and output boxes. In the input box, we specifically annotated
> "1) English Translation." We appreciate your feedback on the clarity of the figure and will work on improving it.
> - Gemini citation fixed (L485)
> - Thank you for your time and effort in helping us improve our presentation.
> - We will ensure that we emphasize how this framework can be applied to three different configurations. (L188-190)

---

### Official Review · Reviewer_qvp3 · 2024-11-04

**Soundness:** 3
**Presentation:** 2
**Contribution:** 3
**Rating:** 6
**Confidence:** 4

**Summary:**

### Summary:
This paper proposes a solution called TransFusion (TF) for Information Extraction (IE) tasks in low-resource languages such as African languages with access to only high-resource (en) IE annotated data. It has two key parts 1/ a simple update to the incumbent model GoLLIE's prompt: At inference time, the low resource text input is translated to English and then the model is instructed to first annotate this english text and then "fuse" the final annotations on the low-resource language input. 2/ Fine-tuning GoLLIE for the above inference. For this, high resource (en) data is used and annotations are projected on the translated text. This gives 4-way tuples of the text as well as annotations in both languages. This is used to finetune GoLLIE to do translate, annotate, fuse.

The paper also discusses training data preparation, the modeling formulation and how an LLM's autoregressive decoding is used to execute on this modeling objective. Finally, results are comprehensively shown on a wide variety of languages and datasets where GoLLIE with TF has distinctly better results. Additionally, the paper also shows a stronger baseline than GoLLIE which first translates the en data to target language(s) and uses that to finetune GoLLIE.

### Overall Recommendation:
My overall recommendation is a soft accept currently. This is mainly because one of the key parts of the finetuning data is currently unclear (as asked in the questions). If that can be simply addressed and/or explained clearly, this paper does have the potential as it solves a wide swath of IE tasks cross lingually.

**Strengths:**

## Quality:
1. Overall the proposal is a simple prompting and Chain-of-thought (CoT) trick. It is clearly defined, motivated and explained from data generation to training fully as seen in Sec 3.1.
2. Clear ablation study
3. Good stronger baseline addition with TransTrain
4. Very extensive experiments, appendix visualizations and results. This also includes manual error analysis.
5. Also shows improvements when built with GPT-4.
6. Good error analysis to the last section of the paper.

## Significance:
This method establishes a clear path for performing information extraction (IE) with generative or encoder-only models, showing a clear tradeoff and presenting easily usable recipes for many low-resource African languages. This would be very impactful for various applications.


## Coherence and Clarity
1. The formulations on page 3 help set expectations on the model before jumping into the LLM prompting. This is very useful to fully understand the proposal.
2. Related work is clearly organized by themes and presented with a comparison to the proposed work.

**Weaknesses:**

1. The novelty of the cross lingual transfer and alignment usage is stated with just 2020+ references. This is overstating the novelty. For example: Structured Prediction as Translation between Augmented Natural Languages (Paolini et al): would be very apt to relate to as it also does IE with generative models.
2. The clarity around the finetuning data formation and some of the formalisms as pointed out in Q1 below is lacking. See questions below for details for where the paper can be improved

**Questions:**

1. Can you clarify the notations on pages 3 and 4? Specifically, on page 3, src refers to english and tgt/trans refers to the low resource language. On page 4, L173, trans refers to the english examples. Is this understanding correct? I believe this is because Page 3 is talking about generating the training data which translates **from** English, while Page 4 talks about the inference time workflows (as used in finetuning) where the input example is translated **to** English.
2. L238-L241 are very confusing. From here, it seems somehow that en data has 19k examples and translated data has another 891. Figure 8 in Appendix also shows the same. However, so far in Section 3.1, L145-151 clearly state that all finetuning/training TF data has to be 4-way parallel. So, what do we mean by just 891 translated examples? Don't the 19k in English also need to have translations and spans mapped as labels?
3. The difference between Trans-Train and GoLLIE-TF is that while both use the same 4-way parallel data, Trans-Train only finetunes in the original GoLLIE style using the translated data pairs (x, y) while GoLLIE-TF uses the 4-way data to finetune the task of translate/annotate/fuse. Is this accurate? If so, can you make this explicit inline?
4. Since you were cost-limited per L229, how did that impact the standard test set sizes that you need to run tests on for a fair comparison with literature?
5. Why do you not compare the fusion step alongside standard rule based aligners?

---

> ### Author Response · Authors · 2024-11-25
>
> Thanks for providing this work and we will include it in the discussion
>
> **Question 1:**
> Thank you for reading our paper in detail. You are correct in your understanding of the notation. We will clarify this in the revised version (L171).
>
> - Page 3: src refers to English, and tgt/trans refers to the translation from English to another language (English → X).
> - Page 4: X refers to a low-resource language, while X-trans refers to the English translation of that low-resource language.
>
> **Question 2:**
>
> To clarify, there are 20,000 examples in total:
> - English: 19,109 examples
> - Translated data: 891 examples
> - The 4-way parallel "translate and project" data corresponds to the 891 examples, which are translated and projected from English.
> - For the remaining 19,109 English examples, we used the original English data without applying transfusion instructions to preserve English task performance.
> - So to summarize, we start from the GoLLIE checkpoint and fine-tune the model on 19,109 English + 891 translated data (Updated in L242)
>
> **Question 3: difference between Trans-Train and GoLLIE-TF **
>
> This understanding is accurate, and we will make the differences clearer in the updated version (L317-319).
>
> **Question 4:**
>
> We initially tested on English data and observed a small F1 difference (1–2 points) between the full test set and the subset of 200 examples. Also, the full test set for MasakhaNER is relatively small (400–500 examples).
>
> **Question 5:**
>
> In Table 5, we compare our results with aligner-based methods such as awesome-align and the state-of-the-art model CoDec (in the "Translate-test" setting) for MasakhaNER.
> For the other IE tasks in Table 2, we did not implement aligner-based methods due to the significant manual effort required to design heuristic rules for each dataset and language. This process is particularly resource-intensive and highly dependent on the specific characteristics of the data.

---

> > ### Comment · Reviewer_qvp3 · 2024-12-02
> > **Reviewed Response**
> >
> > Thanks for taking the time to answer and clarify.
> >
> > - I appreciate the clarifications in text for the formulation.
> > - For your 891 examples and 19109 examples clarification: Since this data is generated from English per L214-216, I am not sure why the entirety of your novel method is limited to 891 examples of finetuning since you can have a lot of English IE data if you don't downsample aggressively.
> > - I am still not sure that the downsizing of the test set to 200 examples is the right place of cost-cutting. The 1-2 points on English can be quite different from the other languages given the proportions of en vs. transfusion training data and 200 is still just about half of the total test set (even if we consider the 400-500 examples). That is a signficant % downsizing.
> >
> > I will keep my scores the same. An NLP venue can be a good option to dive into this work more thoroughly instead of ICLR.

---

### Official Review · Reviewer_TcxU · 2024-11-04

**Soundness:** 3
**Presentation:** 4
**Contribution:** 2
**Rating:** 3
**Confidence:** 4

**Summary:**

This work proposes a simple yet effective approach for improving cross-lingual transfer with a focus on Information Extraction (IE) tasks such as NER, relation extraction, and slot filling. The idea is to leverage an external Machine Translation (MT) system such as NLLB (or any other similar MT system) to translate data from low-resource languages to English, and then leveraging the translation data as additional signal/context for making predictions. The second key ingredient of the approach is the idea of bypassing the alignment step for IE task, where the 'trick' is to first do annotations on the example translated to English before predicting annotations in the target language, where the former seems to be crucial to enable good performance, as verified by an ablation study. In general, this simple idea does provide benefits across a range of IE tasks and languages. Moreover, the authors show the usefulness of the approach on IE tasks with unseen annotation sets, and they show how a similar approach can be applied both to decoder-only models (which are in the main focus of the paper), but also to encoder-style models as well.

**Strengths:**

- A simple idea which is well motivated and well explained the paper, with a good coverage of IE datasets and languages, and good side analyses (e.g., ablation studies, error analyses).
- This idea can be easily combined even with API-gated models without fine-tuning for some quick wins on different IE tasks. Given its simplicity, the entry point to try out the model is quite low (which is a plus).
- The experimental work can be easily reproduced by other researchers.

**Weaknesses:**

- I have concerns about the actual novelty of the work (as advertised in the paper). The use of external MT systems to create some 'silver data' as well as to help with cross-lingual transfer is definitely not a novel idea, and has been tried many times before. We do have many examples of work that explore translation-based cross-lingual transfer for low-resource languages, such as:
-- https://aclanthology.org/2023.emnlp-main.242
-- https://aclanthology.org/2024.naacl-long.298/
-- https://aclanthology.org/2023.emnlp-main.399
- In fact, some of the key baselines are missing from the paper (e.g., improved 'translate-test' baselines as in the aforementioned work)
- Also, it is not new to use external MT systems to enable multilingual instruction-tuning, here are just a small selection of papers:
-- https://aclanthology.org/2024.findings-eacl.90.pdf
-- https://arxiv.org/abs/2407.09879
-- https://aclanthology.org/2024.findings-acl.136.pdf
I would like to see a much broader discussion on how exactly the proposed approach is different and novel here.

- Only one model (GOLLIE-7B) is used as the main baseline model. It would be beneficial to try out the same approach with additional open-weights model (e.g., LLama-3) and also vary the size of the model to verify the impact of model size as well and whether some gains might diminish with larger models. Is it possible to also run some experiments with, say, 70B models from some standard families such as Llama? What about Gemma 2 9B?

- The focus of the work is on IE task simply because of the idea on how to improve the results via generating annotations for the example translated to English first - this seems to me as another hint that this is the only real novelty of the work, as in other 'non-IE' tasks the work would not bring much novelty. I would like to see an extended discussion here. Is the same method applicable to non-IE tasks that require sequence labels such as the 'niche' NLP tasks of POS tagging and dependency parsing? Would the method bring any performance gains to NLU tasks that are typically used to evaluate cross-lingual transfer in some previous research such as NLI on AmericasNLI and XNLI or QA on TydiQA?

**Questions:**

- The main novelty of the work seems to be this idea of generating annotations of the example translated to English before annotating the original sentence (as also discussed under "Weaknesses"). Can the authors comment on the increased cost of this approach and if there is a way to cut the cost of additional generation?

- It is unclear to me what happens if a language is unseen by NLLB. Will the proposed method still work or not (and to what extent)? There were some approaches (e.g., see https://aclanthology.org/2023.emnlp-main.242) that proposed easy adaptations of NLLB to unseen languages - it probably makes sense to try out such approaches as well.

- Given that the evaluation datasets used in this work are quite standard, I would recommend providing also additional reference points (i.e., current state-of-the-art scores) from the literature. This would help the reader to grasp not only the gains over the baseline GOLLIE model (and GPT-4) but also situate the results in a wider context of work on those datasets.

- The authors empirically verify that varying the MT model does not have a profound impact on performance. Can the authors discuss why this observation might hold - is it the inherent limitation of the MT system (e.g., it cannot cover some low-resource languages well regardless of its size) or the inherent advantage of the MT system? Is it also related to simplicity/complexity of the task chosen for this experiment?

- How do language properties (beyond its 'resourceness') affect performance? Can we see some patterns based on language proximity/distance to English?

---

> ### Author Response · Authors · 2024-11-25
>
> **Relation to prior work**
> 1) use silver data to help cross-lingual transfer
>
> - [1] https://aclanthology.org/2023.emnlp-main.242
> - [2] https://aclanthology.org/2024.naacl-long.298/
> - [3] https://aclanthology.org/2023.emnlp-main.399
>
> These studies primarily focus on sentence-level classification tasks such as natural language inference (NLI), where a single label is predicted for an entire sentence. In contrast, our work centers on information extraction (IE) tasks that require annotations at the word- or span-level. These tasks are inherently more complex. For example, leveraging machine translation (MT) for cross-lingual IE data creation is significantly more challenging, as demonstrated in [2], where an additional word aligner was required.
>
> **Improved 'translate-test' baseline**
>
> - multilingual instruction-tuning using MT
> The following papers investigate multilingual instruction tuning by applying MT to translate instruction training data into other languages, which are then blended into training:
> - https://aclanthology.org/2024.findings-eacl.90.pdf
> - https://arxiv.org/abs/2407.09879
> - https://aclanthology.org/2024.findings-acl.136.pdf
>
> However, none of these works focus on IE tasks, which require span-level annotations to be projected from English into other languages. Although this is not the primary contribution of our paper, we combine IE-specific instruction tuning for cross-lingual transfer, addressing unique challenges in this domain.
>
> **Other Open-Weights Models as Baselines (e.g., LLaMA-3 70B or Gemma2 9B)**
>
> While we aim to experiment with larger models, we lack the computational resources to train the base GoLLIE model from scratch. 7B Training requires approximately 50 hours on 2 A100 GPUs. Initial experiments on A40 GPUs took 100 hours but failed to achieve the desired results (e.g., unseen task generalization). This limitation led us to adopt the GoLLIE model as a starting point and propose a lightweight fine-tuning recipe.
>
> **Focus on IE Tasks vs. NLU Tasks**
>
> Our work primarily targets IE tasks. For tasks such as part-of-speech (POS) tagging, prior work suggests that GPT-3.5 already outperforms task-specific models. Similarly, natural language understanding (NLU) tasks (e.g., NLI) do not require span-level annotations, which are the central focus of our proposed method.
>
> **Inference Cost Increase**
>
> Please refer to our discussion in the appendix, Table 7, and Section B ("Limitation"). We provide a detailed comparison of inference times and accuracy.
>
> **Handling Languages Not Covered by NLLB**
>
> For languages not supported by NLLB, we use GPT-3.5 for translation (e.g., Ghomala and Naija). It’s worth noting that NLLB can be replaced by any MT model during inference.
>
> **Impact of Varying MT Model Size**
>
> Our model’s fusion mechanism is robust against errors introduced by MT. As shown in our manual analysis, even when MT makes mistakes, the LLM is able to generate correct results by leveraging the original language data.
>
> **How do language properties affect performance? based on language proximity/distance to English?**
>
> Conneau et al. (2020)[https://aclanthology.org/2020.acl-main.747/] shows the performance generally correlates with the amount of training data available for each language.

---

> > ### Comment · Reviewer_TcxU · 2024-11-26
> > **Response to Authors**
> >
> > Many thanks for sharing the response with some clarifications.
> >
> > Regarding other models: there are actually other models of similar size which could/should have been explored (e.g., Llama-3.1-8B or Gemma-2-9B or smaller variants of these models). An interesting question remains how different model family and size impact performance.
> >
> > Languages not covered by NLLB: what is the 'probability' that they will be well covered by an LLM such as GPT3.5 or Llama 3.1, which, although multilingual, have an even narrower reach than NLLB?
> >
> > I still don't fully buy the argument on novelty through the focus on span-/token-level tasks (which are often easier for transfer methods). The work of https://aclanthology.org/2024.naacl-long.298.pdf does evaluate on MasakhaNER 2.0 (but I acknowledge that it does use a word aligner as well). Again, would a similar method also work for sentence-level tasks? Has someone tried it before?
> >
> > The paper is well written and motivation is fine, but to me its core contribution is a simple technical 'tweak' of the basic idea coming from the EasyProject that was shown to work in practice for span-level/token-level tasks. Moreover, besides my concerns on novelty and impact, I feel that the paper is a better fit for an NLP-first conference (*ACL conferences). While there's merit in the paper, I'll keep my score 'as is' to signal that I don't recommend acceptance at this time.

---

### Meta-Review · Area_Chair_pgAq · 2024-12-16

**Metareview:**

This paper proposes TransFusion, a framework for improving multilingual performance in LLMs by fine-tuning models on English translations of low-resource language data and using "annotation fusion" (i.e., showing a fusion model data in the source language and English and annotating both texts). The experiments use the TransFusion approach to finetune an LLM (GOLLIE-7B) with this setup and evaluate the approach on multiple IE tasks. The results show that this approach improves performance across several IE tasks and languages. The paper also provides many experiments demonstrating the approach's effectiveness beyond the primary task evaluations (ablation, error analysis, and task transfer to unseen label sets).

Strengths:
- The reviewers found the paper to be generally clear and easy to understand (TcxU, qvp3), and the approach is well-motivated (TcxU).
- The paper provides extensive experiments examining how the TransFusion framework works in different settings (TcxU, qvp3, T2WB), and the results and trends are clear and consistent (qvp3, T2WB). The experiments also provide good coverage for many low-resource languages across different datasets (ow8F, T2WB).
- The proposed method is broadly applicable (TcxU, ow8F).

Weaknesses:
The primary weakness of this work is a lack of novelty compared to the proposed methods and findings in prior works. Specifically, the components combined into this framework have already been proposed for improving multilingual prompting in prior work (as raised by TcxU, ow8F, and T2WB). The paper does not clearly differentiate why this approach is significantly different from these prior works and misses some related work that previously used these methods (TcxU). Notably, a very related baseline (translate-test) is not included for the full set of results (TcxU, T2WB) - only for a subsequent results table (Table 5) on a subset of tasks.

Some other minor weaknesses were raised by the reviewers, such as that it is unclear how effective the method will be beyond IE tasks (TcxU) and that some points within the paper were not described clearly (qvp3, ow8F). The authors fully addressed other weaknesses and points of confusion during the discussion period.

**Additional Comments On Reviewer Discussion:**

The authors responded to each reviewer; however, the response was not always comprehensive and missed some points raised by the reviewers. One reviewer increased their score after the response (T2WB). Multiple reviewers think this paper would be a better fit for an NLP venue than ICLR (TcxU, qvp3).

---

### Decision · Program_Chairs · 2025-01-22

Reject